# Maxwell Electrodynamics in Terms of Physical Potentials

**Parthasarathi Majumdar [1],\* and Anarya Ray [2]**

[1]   Indian Association for the Cultivation of Science, Kolkata 700032, India
[2]   Department of Physics, Presidency University, Kolkata 700032, India
\*   Correspondence: bhpartha@gmail.com

**Abstract:** A fully relativistically covariant and manifestly gauge-invariant formulation of classical Maxwell electrodynamics is presented, purely in terms of gauge-invariant potentials without entailing any gauge fixing. We show that the inhomogeneous equations satisfied by the physical scalar and vector potentials (originally discovered by Maxwell) have the same symmetry as the isometry of Minkowski spacetime, thereby reproducing Einstein's incipient approach leading to his discovery of special relativity as a spacetime symmetry. To arrive at this conclusion, we show how the Maxwell equations for the potentials follow from stationary electromagnetism by replacing the Laplacian operator with the d'Alembertian operator, while making all variables dependent on space and time. We also establish consistency of these equations by deriving them from the standard Maxwell equations for the field strengths, showing that there is a unique projection operator which projects onto the physical potentials. Properties of the physical potentials are elaborated through their iterative Nöther coupling to a charged scalar field leading to the Abelian Higgs model, and through a sketch of the Aharonov–Bohm effect, where dependence of the Aharonov–Bohm phase on the physical vector potential is highlighted.

## 1. Introduction

The standard textbook formulation of Maxwell electrodynamics, in vacua with sources, entails linear first order partial differential equations for electric and magnetic field strengths $\vec{E}$ and $\vec{B}$. Conventionally, the equations for these field strengths are first cast in terms of the scalar and vector *potentials*, $\phi$ and $\vec{A}$. The resulting second order equations for the potentials are found to be noninvertible because of the *gauge ambiguity* of the potentials—addition of gradients of arbitrary (gauge) functions to any solution generates an equivalence class of solutions for the potentials, related by local gauge transformations. All the gauge potentials in a gauge-equivalent class give the same electromagnetic field strengths. This gauge ambiguity has often led people to consider gauge potentials as *unphysical*, in comparison to the 'physical' (gauge-invariant) field strengths. The standard procedure for getting to the solutions is to 'gauge fix' the potentials, i.e., impose subsidiary conditions on them so that the ambiguity may be resolved. There is a nondenumerably infinite set of such subsidiary 'gauge conditions', each one as ad hoc as the other, and none with any intrinsic physical relevance. This entire approach, tenuous as it is, avoids facing up to the central issue: Why are the equations for the potentials noninvertible in the first place? Is Nature so unkind as to provide us with unique gauge-invariant equations for quantities which themselves are infinitely ambiguous? The answer is an emphatic *No!*

In his succinctly beautiful history of the Maxwell equations of electrodynamics, Nobel Laureate theoretical physicist C. N. Yang [1] recalls how Faraday first identified the concept of the 'electrotonic state' as the origin of the induced electromotive force, purely as a result of his extraordinary experimental research and physical intuition. The idea of the vector potential was introduced by Thomson (Lord Kelvin) in 1851, ostensibly as a solution of $\nabla \cdot \vec{B} = 0$. Five years later, in a brilliant

identification of Thomson's vector potential with Faraday's electrotonic state, Maxwell wrote down, for the first time ever, the equation $\vec{E} = -\dot{\vec{A}}$, which led him to his Law VI: *The electromotive force on any element of a conductor is measured by the instantaneous rate of change of the electrotonic intensity on that element, whether in magnitude or direction.* Yang further writes: "The identification of Faraday's elusive idea of the electrotonic state (or electrotonic intensity, or electrotonic function) with Thomson's vector potential is, in my opinion, the first great conceptual breakthrough in Maxwell's scientific research...", also, "Indeed, the concept of the vector potential remained central in Maxwell's thinking throughout his life."

From our standpoint, it is inconceivable that such an outstanding experimentalist as Faraday would focus on a concept which we call the vector potential, if indeed it is 'unphysical', as often perceived nowadays among a certain group of physicists. Likewise, Maxwell's preoccupation with the same concept would have been a continuation of an illusory pursuit if the vector potential is indeed unphysical. Interestingly, Maxwell himself was apparently quite aware of the gauge ambiguity of the Maxwell equations for $\vec{E}$ and $\vec{B}$ as mentioned above, but according to Yang [1], on the issue of gauge-fixing Maxwell was silent: 'He did not touch on that question, but left it completely indeterminate.' This is where we speculate on the reason: Maxwell was perhaps aware that his equations themselves provided, in today's parlance, a *unique projection operator* which projects onto the physical part of the vector potential. Clearly, $\vec{E}$ and $\vec{B}$ depend *solely on this physical part of the vector potential* and are quite independent of the unphysical *pure gauge* part. The manner in which the projection operator isolates the gauge-invariant physical part of the vector potential, can, of course be reproduced by a gauge choice as well—however, such a choice is *by no means* essential. Gauge choices (or gauge-fixings) merely constrain the unphysical part of the gauge potential, leaving the gauge-invariant physical part quite untouched, as they must.

To reiterate, the reason that the equations for the potentials are noninvertible in the first place is because their intrinsic analytic structure involves a *projection* operator which has a nontrivial kernel of unphysical 'pure gauge' vector fields! This simple observation renders any 'gauge fixing' superfluous, since it is now obvious that the equations are to be interpreted in terms of projected *physical, gauge-invariant* potentials not belonging to the kernel of the projection operator, hence obeying very simple wave equations that are immediately uniquely invertible without the need for any imposition of additional 'gauge conditions'. We find it surprising that this simple fact has not been clarified in any of the number of currently popular textbooks on classical electrodynamics. From a physical standpoint, this approach, in contrast to the standard one based on the field strengths, immediately divulges the essence of electromagnetism as the theory of electromagnetic *waves* under various circumstances. All other field configurations (in vacuo) can be easily explained once the propagation and generation of electromagnetic waves is understood in terms of the physical potentials.

There is another lacuna in extant textbook treatments of Maxwell electrodynamics—the absence of a fully relativistically covariant formulation of the subject ab initio. Special relativity is intrinsically embedded in Maxwell electrodynamics with charge and current sources in empty space, as was discovered by Einstein in 1905 [2]. If, as per standard practice, the fundamental equations are written in terms of the electric and magnetic fields, the relativistic invariance of these equations is far from obvious. This emerges only after some effort is given to relate the electric and magnetic fields in different inertial reference frames connected by Lorentz boosts. In contrast, if the equations are cast in terms of the *physical* electromagnetic scalar and vector potentials introduced by Maxwell, then these potentials and the equations they obey can be easily combined to yield a structure that is *manifestly* invariant under Lorentz boosts as well as spatial rotations, i.e., the full Lorentz transformations. Given that it is easier always to compute four, rather than six, field components for given source charge and current densities, it stands to reason to begin any formulation of electrodynamics from the (physical) potentials, rather than the field strengths.

Despite its antiquity, a formulation of classical electrodynamics that, from the outset, is fully relativistically covariant, is somehow not preferred in the very large number of excellent textbooks

currently popular, with perhaps the sole exception of [3]. Even so, the issue of the gauge ambiguity and the full use of electromagnetic potentials rather than fields has not been dealt with adequately, even in this classic textbook. Thus, while relativistically covariant Lienard–Wiechert potentials describing the solution of Maxwell's equations due to an arbitrarily moving relativistic point charge have been obtained, the corresponding field strengths and the radiative energy–momentum tensor have not been given such a manifestly covariant treatment. The more widely used textbook [4] also fills this gap only in part. Since special relativity is so intrinsic to Maxwell vacuum electrodynamics with sources, it is only befitting that the entire formalism exhibit this symmetry explicitly. The subtle interplay with gauge invariance is also a hallmark of this theory, which forms the basis of our current understanding of the fundamental interactions of physics.

We end this introduction with the disclaimer—this paper is exclusively on Maxwell electrodynamics. As such, it does not discuss theoretically interesting generalizations involving magnetic charges and large gauge transformations. Interesting as these ideas are, there is no observational evidence yet that they are applicable to the physical universe, so these ideas remain within the domain of speculation. Of course, the moment a magnetic monopole is observed as an asymptotic state, our paper stands to be immediately falsified. This, however, is not a lacuna of the paper, rather it is its strength that it 'sticks its neck out' so to speak, in contrast with the plethora of theoretical papers whose veracity or relevance vis-a-vis the physical universe remains forever in doubt. Regarding the generalization of electrodynamics with both electric and magnetic charges, the construction of a local field theory is still not without issues. Whether this is a hint from Nature about the relevance of such ideas, still remains unclear. In our favor, the entire description being in terms of a single 4-vector potential has a virtue: The electric and magnetic aspects are actually *unified* in this description. If magnetic monopoles were present, this unification would actually be absent, in favor of a duality symmetry—the electric–magnetic duality. It is not unlikely that even though this duality symmetry is dear to some theoretical physicists, Nature does not make use of it, if the evidence so far is to be taken into account. In regard to large gauge transformations, recent work on asymptotic symmetries of flat spacetime has led to interesting issues regarding electrodynamic gauge transformations, which may serve as research topics for the future.

This paper is structured as follows: In Section 2 we first exhibit gauge-invariant physical potentials for stationary electromagnetism (electrostatics and magnetostatics) and show how they satisfy *identical* equations underlining their inherent unity. We then generalize this to full electrodynamics with a neat substitution, and show that the standard Maxwell equations for field strengths *emerge* from these. In the next section, we demonstrate the invariance of the Maxwell equations for the physical potentials under Lorentz transformations, characterized by a 4 × 4 matrix $\Lambda$ which includes both spatial rotations and Lorentz boosts. We argue that this symmetry of Maxwell electrodynamics is also the isometry of the Minkowski metric of flat spacetime. Next, we complete the circle by showing how the equations for the physical potentials can be derived covariantly from the covariant Maxwell equations for the field strengths and exhibit the form of the projection operator, which enables this projection onto physical potentials. Then, in Section 4, we show how the physical potentials can be coupled gauge-invariantly to charged scalar fields through the technique of *iterative Nöther coupling*, leading to the classical Abelian Higgs model. We also provide a sketch to show that the Aharonov–Bohm phase is a functional of the physical potentials alone, completely independent of the unphysical pure gauge part. We conclude in Section 5.

## 2. Gauge-Invariant Physical Potentials

### 2.1. Stationary Electromagnetism and Physical Potentials

Electrostatics is described by the equations

$$
\begin{aligned}
\nabla \cdot \vec{E} &= \rho, \\
\nabla \times \vec{E} &= 0,
\end{aligned}
\tag{1}
$$

where constants appropriate to choice of units have been set to unity. The solution of the second line of Equation (1) is $E = -\nabla\phi$, where $\phi$ is the scalar potential. $\phi$ is unique modulo an added constant. Substitution of this solution into the first line of Equation (1) results in the Poisson equation for $\phi$:

$$\nabla^2 \phi_P = -\rho. \tag{2}$$

Here, we have added a subscript $P$ to $\phi$ to emphasize its physicality. Equation (2) has the (inhomogeneous) solution:

$$\phi_P(\vec{x}) = \int d^3x' \, \frac{\rho(\vec{x}')}{|\vec{x} - \vec{x}'|}. \tag{3}$$

Likewise, stationary magnetism begins with the equation:

$$\begin{aligned} \nabla \cdot \vec{B} &= 0, \\ \nabla \times \vec{B} &= \vec{j}. \end{aligned} \tag{4}$$

Apparently, the equations in (4) have nothing in common with their electrostatic counterpart (1), leading to the idea espoused in textbooks that electrostatics and magnetostatics are two separate disciplines. We now show that this idea is not correct at all, once the equations are transcribed in terms of the *physical* scalar and vector potentials.

Solving the first of the Equation (4) in terms of the vector potential $\vec{A} : \vec{B} = \nabla \times \vec{A}$, and substituting that in the second line of (4) yields

$$\nabla(\nabla \cdot \vec{A}) - \nabla^2 \vec{A} = \vec{j}. \tag{5}$$

Now, Equation (5) cannot be solved uniquely for $\vec{A}$ because of the gauge ambiguity—there is an infinity of *gauge-equivalent* solutions for any $\vec{A}$ that satisfies (5). Usually, the way out of this ambiguity, as mentioned in almost all textbooks, is *gauge fixing*—choose $\nabla \cdot \vec{A}$ to be any specific function $f(\vec{x})$, leading to the vectorial Poisson equation which can be solved immediately with suitable boundary conditions, for every given source $\vec{j}$. The choice $f(\vec{x}) = 0$ is called the Coulomb gauge in some textbooks. But this is *not* what we propose to do here!

Instead, consider the Fourier transform of (5):

$$-\vec{k}[\vec{k} \cdot \vec{A}(\vec{k})] + k^2 \vec{A}(\vec{k}) = \vec{j}(\vec{k}), \tag{6}$$

where $k^2 = k_a k_a$, $a = 1, 2, 3$. Switching to component notation, Equation (6) can be rewritten as

$$k^2 \, \mathcal{P}_{ab} \tilde{A}_b = \tilde{j}_b, \tag{7}$$

where the *projection operator* $\mathcal{P}_{ab}$ is defined as

$$\mathcal{P}_{ab} \equiv \delta_{ab} - \frac{k_a k_b}{k^2}, \quad \mathcal{P}_{ab} \, \mathcal{P}_{bc} = \mathcal{P}_{ac}. \tag{8}$$

Clearly, in defining this projection operator, we have chosen $k^2 \neq 0$, i.e., $\vec{k}$ is a nonzero vector. We shall comment on the zero vector situation below.

Define now the projected vector potential $\vec{A}_P$ through the equation

$$\tilde{A}_{Pa} \equiv \mathcal{P}_{ab} \, \tilde{A}_b. \tag{9}$$

This projected vector potential satisfies two very important properties: First of all, it is *gauge-invariant* and hence, *physical*! This follows from the fact that $k_a \mathcal{P}_{ab} = 0 \ \forall \ \vec{k} \neq 0$. Secondly, this last relation also implies that the projected vector potential satisfies

$$\vec{k} \cdot \vec{A}_P = 0 \Rightarrow \nabla \cdot \vec{A}_P = 0 \tag{10}$$

*automatically, without having to make any gauge choice*! Note that this is *not* the so-called Coulomb gauge choice. It is rather the consequence of defining the projected vector potential $A_P$ using the projection operator that occurs already in Maxwell's equations for magnetostatics. No extraneous choice needs to be made for this physical projection—it is a unique projection. In fact, it now becomes clear as to why Ampere's law, written out in terms of the full vector potential, is not invertible: This equation involves the projection operator $\mathcal{P}_{ab}$, projection operators are not invertible because they have a nontrivial kernel—here, it is the set of pure gauge configurations expressible as $\nabla a$ for arbitrary scalar functions $a$. Once the projected vector potential $A_P$ is defined, it satisfies the vector Poisson equation

$$\nabla^2 \vec{A}_P = -\vec{j}, \tag{11}$$

with the solution (in Fourier space)

$$\vec{A}_P = \frac{\vec{j}}{k^2} . \tag{12}$$

In position space, this solution is

$$\vec{A}_P(\vec{x}) = \int d^3 x' \frac{\vec{j}(\vec{x}')}{|\vec{x} - \vec{x}'|}, \tag{13}$$

which clearly shows that, unlike the magnetic field, the vector potential tracks the current producing it.

There is an issue of a residual gauge invariance for $k^2 = 0$ which has been excluded from our earlier discussion. If $k^2 = 0$, in position space, this would be taken to imply that the projected vector $\vec{A}_P$ has the residual ambiguity under the gauge transformation $\vec{A}_P \rightarrow \vec{A}_P + \nabla \omega$ with $\nabla^2 \omega = 0$. However, from the uniqueness of solutions of Laplace equation, we know that choosing the boundary condition $\omega = constant$ at spatial infinity implies that $\omega = const$ everywhere, thereby precluding any such residual ambiguity for the physical $\vec{A}_P$.

Summarizing stationary electromagnetism, we have, as the fundamental equations in terms of the physical potentials,

$$\nabla^2 \phi_P = -\rho. \tag{14}$$
$$\nabla^2 \vec{A}_P = -\vec{j}. \tag{15}$$
$$\nabla \cdot \vec{A}_P = 0 . \tag{16}$$

The great mathematical unity between the physical electric potential and magnetic vector potentials, in terms of the equations they satisfy, need hardly be overemphasized. It is also perhaps the most succinct manner of presentation of stationary electromagnestism.

### 2.2. Formulation of Full Electrodynamics in Terms of Physical Potentials

The passage from the stationary Equations (14)–(16) to the full time-dependent equations for the physical potentials follows the exceedingly simple rule: Allow all functions of space to now be functions of space and time, i.e., $\vec{x}$ and $t$—and make the simple change $\nabla^2 \rightarrow \Box^2$ in Equations (14) and (15), where, $\Box^2 \equiv \nabla^2 - (\partial^2/\partial t^2)$ is the d'Alembertian operator, and we are using units such that $c = 1$. Further, the divergence-free condition (16) is to be augmented by the 'spacetime' divergence-free

condition $(\partial \phi_P / \partial t) + \nabla \cdot \vec{A} = 0$, leading to the Maxwell equations for the physical electromagnetic scalar and vector potentials:

$$
\begin{aligned}
\Box^2 \phi_P &= -\rho, \\
\Box^2 \vec{A}_P &= -\vec{j}, \\
\frac{\partial \phi_P}{\partial t} &+ \nabla . \vec{A}_P = 0,
\end{aligned}
\tag{17}
$$

where all other constants are absorbed into redefinitions of $\rho$ and $\vec{j}$.

Equation (17) can be motivated physically ab initio from the most important characteristic of electrodynamics, namely, that they must yield electromagnetic waves traveling through empty space. Indeed, the top two equations in (17) are nothing but *inhomogeneous* wave equations, recalling that the d'Alembertian is the *wave* operator. The last equation is a special characteristic property of the physical potentials $\phi_P$, $\vec{A}_P$, which is relevant to ensure that the electromagnetic waves in empty space have transverse polarization. We think that electromagnetic waves constitute the most important physical property of electrodynamics, and our formulation of Maxwell's theory in terms of the potentials brings out this characteristic immediately and without the need for extraneous manipulations. Thus, one can start with the formulation in terms of Equation (17) by pointing out that they epitomize electromagnetic waves, and form the basis of what we see, of ourselves, of worlds outside ours and also in terms of constructing theories of fundamental interactions.

The standard Maxwell equations for the electric and magnetic field strengths, $\vec{E}$ and $\vec{B}$, result from (17) immediately upon using Maxwell's definition of Faraday's 'electrotonic' state, as defined in the introduction: $\vec{E} \equiv -\partial \vec{A}_P / \partial t - \nabla \phi_P$ and Thomson's definition $\vec{B} \equiv \nabla \times \vec{A}_P$. One obtains

$$
\begin{aligned}
\nabla \cdot \vec{E} &= \rho, \\
\nabla \cdot \vec{B} &= 0, \\
\nabla \times \vec{E} &= -\frac{\partial \vec{B}}{\partial t}, \\
\nabla \times \vec{B} &= \vec{j} + \frac{\partial \vec{E}}{\partial t}.
\end{aligned}
\tag{18}
$$

## 3. Special Relativity as a Symmetry of Maxwell's Equations

### 3.1. Manifest Lorentz Symmetry

Observe that these equations can be combined into the 4-vector potential $\mathbf{A}_P$ with components $A_P^\mu, \mu = 0, 1, 2, 3$, with $A_P^0 = -A_{P0} = \phi$, and $A_P^m = A_{P,m}, m = 1, 2, 3$, where $\vec{A}_P = \{A_P^m | m = 1, 2, 3\}$. Similarly, the charge and current densities can be combined into a current density 4-vector $\mathbf{J}$ with components $J^\mu \ (\mu = 0, 1, 2, 3) = (\{\rho, j^a\}, a = 1, 2, 3)$.

Writing $\partial_\mu \equiv \partial / \partial x^\mu$, Equation (17) can now be summarized as

$$
\begin{aligned}
\Box^2 A_P^\mu &= -J^\mu, \\
\partial_\mu A_P^\mu &= 0.
\end{aligned}
\tag{19}
$$

Raising and lowering of indices are effected by the Minkowski spacetime invariant metric tensor $\eta_{\mu\nu} = \eta^{\mu\nu} = diag(-1, 1, 1, 1)$.

It is to be noted that Equations (17) and (19) are *not* gauge-fixed versions of equations for potentials corresponding to standard Maxwell equations, even though they look enticingly similar. In other words, the second of the equations in (19) is *not a gauge choice*, but a compulsion from Nature. We shall make this clear shortly. Thus, these equations are to be treated at the same physical footing as the standard Maxwell equations for the field strengths, containing the same physical information as the latter, without any ambiguity.

Observe now that the d'Alembertian operator $\Box^2 \equiv \eta^{\mu\nu}\partial_\mu\partial_\nu$ is *invariant* under the transformation $\partial_\mu \to \partial'_\mu = \Lambda_\mu{}^\nu\partial_\nu$, provided the transformation matrix $\Lambda$ satisfies

$$\eta_{\mu\nu}\,\Lambda^\mu{}_\rho\,\Lambda^\nu{}_\sigma = \eta_{\rho\sigma}\,. \tag{20}$$

It follows that both lines of Equation (19) are invariant under these transformations, if $A'^\mu(x') = \Lambda^\mu{}_\nu A^\nu(x)$, $J'^\mu(x') = \Lambda^\mu{}_\nu J^\nu(x)$, and $x'^\mu = \Lambda^\mu{}_\nu x^\nu$, with the transformation matrices $\Lambda^\mu{}_\nu$ satisfying (20). As is expected, the coordinate transformations leave invariant the squared invariant interval in Minkowski spacetime $ds^2 = \eta_{\mu\nu}\,dx^\mu\,dx^\nu$. Thus, the transformations that leave the equations of electrodynamics invariant are precisely the same transformations that constitute a *symmetry* of Minkowski spacetime. It is obvious that these are the full *Lorentz* transformations, including spatial rotations and Lorentz boosts, e.g., if $\Lambda^0{}_0 = 1$, $\Lambda^0{}_m = 0$, the remaining $3 \times 3$ submatrix constitutes the *orthogonal* transformation matrix corresponding to rotations in 3-space. Likewise, if $\Lambda^0{}_0 = \gamma = \Lambda^1{}_1$; $\Lambda^0{}_1 = -\beta\gamma = \Lambda^1{}_0$ etc., that constitutes a Lorentz boost in the $+x^1$ direction. The Lorentz factor $\gamma = (1 - \beta^2)^{-1/2}$. Thus, *all* Lorentz boosts and spatial rotations are just choices for the $\Lambda$ matrix subject to the restriction (20).

The standard Lorentz-covariant equations of vacuum electrodynamics involving field strengths are easily obtained from Equation (19) upon using the standard definition $F_{\mu\nu} \equiv \partial_\mu A_{P\nu} - \partial_\nu A_{P\mu}$, leading immediately to the transformation law under the $\Lambda$-transformations: $F^{(\Lambda)}_{\mu\nu} = \Lambda^\rho_\mu \Lambda^\sigma_\nu\,F_{\rho\sigma}$, and the equations

$$
\begin{aligned}
\partial_\mu F^{\mu\nu} &= -J^\nu, \\
\partial_\mu(e^{\mu\nu\rho\sigma}\,F_{\rho\sigma}) &= 0\,.
\end{aligned}
\tag{21}
$$

We acknowledge the influence of the Feynman lectures on physics [5] in basing the formulation presented above on the potentials rather than the fields. However, the delineation of the central role of the *projection operator* inherent in the Maxwell equations, for stationary electromagnetism above, as also for the full theory, to be given in the subsection to follow, is original to the best of our knowledge.

### 3.2. Closing the Circle: Physical Vector Potentials from the Standard Formulation

### 3.2.1. With Sources

We begin by defining the field strengths $F_{\mu\nu}$ in terms of the standard *gauge potential* $A_\mu$ (i.e., without the subscript '$P$'): $F_{\mu\nu} \equiv \partial_\mu A_\nu - \partial_\mu A_\nu$. The second of the standard Maxwell equations (21) results immediately. The first of (21) is then either postulated on the basis of experiments, or derived from the Maxwell action [3]. Be that as it may, one may substitute the above definition of the field strengths $F_{\mu\nu}$ in terms of the gauge potentials, yielding

$$\partial_\mu\partial^\mu A^\nu - \partial^\nu\partial_\mu A^\mu = -J^\nu, \tag{22}$$

which, under Fourier transformation (in four dimensions) with Fourier variable $k^i$, $i = 0,1,2,3$, leads to the equation

$$-k^2\,\mathcal{P}^\nu_\mu\,\tilde{A}(k)^\mu = \tilde{J}^\nu(k). \tag{23}$$

$$\mathcal{P}^\nu_\mu \equiv \delta^\nu_\mu + \frac{k^\nu k_\mu}{k^2}\,,\ k^2 \equiv k_\rho k^\rho \neq 0\,. \tag{24}$$

We first confine to the inhomogeneous Maxwell equation, and take up the homogeneous case later. The projection operator $\mathcal{P}^\nu_\mu$ above possesses the properties characteristic of projection operators in general.

$$\mathcal{P}^\mu_\nu \, \mathcal{P}^\nu_\rho \;=\; \mathcal{P}^\mu_\rho. \tag{25}$$

$$\mathcal{P}^\mu_\nu \, k^\nu \tilde{f}(k) \;=\; 0 \,\forall\, \tilde{f}(k). \tag{26}$$

where (26) characterizes the vectors in the kernel of the projection operator.

The fact that the vacuum Maxwell equation with sources is expressed uniquely and naturally in terms of a *projection* of the gauge potential, without having to make any choices, is of crucial importance, since the projection is clearly on the gauge-invariant physical subspace. This projected vector potential, defined as $A_{P\mu} \equiv \mathcal{P}^\nu_\mu A_\nu$, has the following essential properties, which can be easily gleaned from Fourier space: (a) $\partial_\mu A^\mu_P = 0$, i.e., it is spacetime transverse; (b) under gauge transformations $A_\mu \to A^{(\omega)}_\mu = A_\mu + \partial_\mu \omega$, the projected (physical) vector potential $A^{(\omega)}_{P\mu} = A_{P\mu}$, i.e., it is gauge-invariant and hence, *physical*! This implies that

$$A_\mu = A_{P\mu} + \partial_\mu a, \tag{27}$$

so that the entire burden of gauge transformations of $A_i$ is carried by $a(x) : A_\mu \to A_\mu + \partial_\mu \omega \Rightarrow a^{(\omega)} = a + \omega$, which underlines the complete unphysicality of the pure gauge part ('longitudinal' mode) $a$ of $A$. It also follows trivially that $F_{\mu\nu}(A) = F_{\mu\nu}(A_P)$, which means that invariance under gauge transformations does not represent a physical symmetry, but merely a *redundancy* in the gauge potential [6]. One also sees that Equation (19) results immediately from our consideration, so we have come full circle. In fact, in Fourier space, we have an explicit solution for the physical potential $A_{P\mu}$ in terms of the sources:

$$\tilde{A}_{P\mu}(k) = -\frac{\tilde{J}_\mu(k)}{k^2} \tag{28}$$

so that, given the form of the 4-vector source, the physical potential and field strengths are determined in spacetime through appropriate inverse Fourier transforms.

In our proof of the gauge invariance of the projected 4-vector potential $\mathbf{A}_P$ above, the special case of gauge functions $\omega$ satisfying $\Box^2 \omega = 0$ has been excluded. We notice that if such gauge functions are retained, the projected 4-vector potential is seen have a *residual* gauge ambiguity involving the spacetime gradient of such gauge functions. This residual ambiguity arises even if we impose the Lorentz–Landau gauge as in the standard textbooks. Note that this residual ambiguity may be eliminated, as in standard procedure, if we restrict our attention to electromagnetic field strengths that decay to vanishingly small values at infinity. This implies that the projected physical potential must vanish at infinity as well, leaving the longitudinal mode $a$ to turn into a constant at most at infinity. This implies that the only gauge transformations that are permitted at infinity and solve the wave equation, are constants, whose gradients vanish everywhere. Thus, just as with stationary magnetic fields, the residual gauge ambiguity is no cause for concern.

### 3.2.2. Without Sources

Consider now the homogeneous or *null* case, when $J^\mu = 0$ in Equation (22). In this case, if $k^2 \neq 0$, the projected vector potential, which is proved to be physical and gauge-invariant above, must vanish, leaving only the unphysical pure gauge part which leads to vanishing field strengths. That solution is devoid of any physical interest.

Thus, it is obvious that for nontrivial electromagnetic fields, $k^2 = 0$, i.e., **k** is a null spacetime vector. In this case, it also follows from the Fourier transformed version of (22) that, for every nontrivial null vector **k**, we must have

$$\mathbf{k} \cdot \mathbf{A} = 0 \Rightarrow \partial_\mu A^\mu = 0. \tag{29}$$

Observe that this is not a choice, but simply follows from the standard Maxwell equations written out in terms of the vector potential.

We recall that 4-dimensional spacetime $\mathcal{M}(3,1)$ can be represented at every point as the Cartesian product $\mathcal{M}(1,1) \times \mathbf{R}^2$, where the first factor is two-dimensional Lorentzian spacetime, and the second is just the Euclidean plane [7]. With this, we realize that there is another null vector **n** linearly independent of **k**, which, together with **k**, spans $\mathcal{M}(1,1)$. One can always choose **n** such that $\mathbf{n} \cdot \mathbf{k} = -1$ for our signature of the Lorentzian metric.

Now, what is known about propagation of electromagnetic waves in vacuum [3,4] is that these waves are endowed with *transverse* spatial polarization, i.e., the electric field (and hence the 3-vector potential) must oscillate in a plane transverse to the spatial direction of propagation. This implies that, for freely propagating electromagnetic waves in vacua, the 4-vector potential cannot possibly have any component in the direction of propagation of light in *spacetime*, thereby precluding any components tangent to $\mathcal{M}(1,1)$. It can have only two *spacelike* physical components, both lying in the Euclidean plane $\mathbf{R}^2$. These physical requirements must be encoded in the projection operator for the source-free case.

It follows that the physical vector potential, defined as

$$A_{P\mu} \equiv \mathcal{P}_{\mu\nu} A^\nu , \ \mathcal{P}_{\mu\nu} \equiv \eta_{\mu\nu} + k_\mu n_\nu + n_\nu k_\mu , \tag{30}$$

has the properties of being transverse to $\mathcal{M}(1,1)$, having two components both of which are tangential to the Euclidean plane, and is also *gauge-invariant*.

$$\mathbf{k} \cdot \mathbf{A}_P = 0 = \mathbf{n} \cdot \mathbf{A}_P . \tag{31}$$

The latter property follows from the uniqueness of solutions of the two-dimensional Laplace equation. Thus, the gauge-invariant, physical components of the 4-potential satisfying the homogeneous wave equation have their polarization vectors pointing in the two linearly-independent directions of the Euclidean plane $\mathbf{R}_2$. It follows that the projected physical 4-potential $A_P$ is *spacelike* in character. Thus, gauge-invariance is at the root of transverse polarization of electromagnetic waves in vacuum.

## 4. Applications of the Physical Potentials

### 4.1. Coupling to Charged Scalar Fields

The action for a self-interacting charged scalar field $\psi$ is, in general, given by the action

$$S_0[\psi] = \int d^4x [\partial_\mu \psi (\partial^\mu \psi)^* - V(|\psi|)] . \tag{32}$$

Using the radial decomposition $\psi = (1/\sqrt{2})\rho(x) \exp i\Theta(x)$, this action is rewritten as

$$S_0[\rho, \Theta] = \int d^4x [\frac{1}{2}\partial_\mu \rho \partial^\mu \rho + \frac{1}{2}\rho^2 \partial_\mu \Theta \partial^\mu \Theta - V(\rho)] . \tag{33}$$

This action is clearly invariant under the *global* $U(1)$ transformation $\rho \to \rho$ , $\Theta \to \Theta + \omega$, for a constant real parameter $\omega$. The conserved Nöther current corresponding to this global symmetry is given by $J^\mu = \rho^2 \partial^\mu \Theta$.

We now add the Maxwell action $S_{Max}[A_P] = -(1/4) \int d^4x F^{\mu\nu}(A_P)F_{\mu\nu}(A_P)$ to $S_0$. The coupling of the charged scalar field to the physical potential is now affected through the interaction term $S_1 = \int d^4x J_\mu A_P^\mu$, leading to the full action $S[\rho, \Theta, A_P] = S_0 + S_{Max} + S_1$. It is obvious that $S$ is also symmetric under global $U(1)$ transformation of the $\Theta$ field, with both $\rho, A_P$ remaining invariant. However, because of the additional interaction term $S_1$, the conserved Nöther current corresponding to the global symmetry is now augmented to $J'_\mu = \rho^2(\partial_\mu\Theta + A_{P\mu})$. Following the prescription of the iterative Nöther coupling [8], we now couple this augmented current to the physical potential $A_P$, so as to obtain the full action

$$S'[\rho, \Theta, A_P] = \int d^4x[-\frac{1}{4}F_{\mu\nu}F^{\mu\nu} + \frac{1}{2}\partial_\mu\rho\partial^\mu\rho + \frac{1}{2}\rho^2(D_\mu\Theta D^\mu\Theta) - V(\rho)] \,, \tag{34}$$

where $D_\mu\Theta \equiv \partial_\mu\Theta + A_{P\mu}$. This action is clearly invariant under global $U(1)$ symmetry transformations. Further iterations of the Nöther current interaction leads to no new terms in the action [8].

If this action is rewritten in terms of the *full* gauge potential $A^\mu \equiv A_P^\mu + \partial^\mu a$, with $a(x) \equiv \int d^4x \mathcal{G}(x-x')\partial'_\mu A^\mu(x')$, $\Box^2\mathcal{G}(x-x') = \delta^{(4)}(x-x')$, and we introduce a new field $\chi \equiv \Theta - a$, then $D_\mu\Theta = \partial_\mu(\chi + a) + (A_\mu - \partial_\mu a) = \partial_\mu\chi + A_\mu \equiv D_\mu\chi$. Writing $\phi = (1/\sqrt{2})\rho \, \exp i\chi$, we obtain the net action

$$S'[\phi, A] = \int d^4x[D_\mu\phi(D^\mu\phi)^* - V(|\phi|) - \frac{1}{4}F_{\mu\nu}(A)F^{\mu\nu}(A)], \tag{35}$$

where $D_\mu\phi = \partial_\mu\phi + iA_\mu\phi$. This action is clearly invariant under *local* $U(1)$ gauge transformations: $\phi \to \phi \exp i\omega(x)$, $A_\mu \to A_\mu - \partial_\mu\omega$. Thus, the iterative Nöther coupling prescription leads uniquely to the $U(1)$ gauge-invariant action for the charged scalar field (35). The starting point is of course the physical vector potential $A_{P\mu}$. In terms of the fields $\rho, \chi, A_P$, and $a$, the local $U(1)$ gauge transformations do not affect $\rho, A_P$ but only $\chi \to \chi + \omega$, $a \to a - \omega$. The minimal coupling prescription is not used here, but emerges from the prescription of iterative Nöther coupling.

### 4.2. Aharonov–Bohm Effect

The Aharonov–Bohm effect [9] is historically the first tested proposal to underline the physicality of magnetic vector potential $\vec{A}_P$. This effect is a quantum mechanical effect which shows that the wave function of an electron in a closed orbit in a classical vector potential (even if there is no magnetic field in the region) will pick up a geometric phase given by the anholonomy of the vector potential along the closed curve. This phase is thus given by the expression $\Phi(\vec{A}) = \int_C \vec{A} \cdot d\vec{l}$, where $C$ is a noncontractible loop. Now, in our approach, the vector potential admits the decomposition $\vec{A} = \vec{A}_P + \vec{A}_U$, where $\vec{A}_U = \nabla a(\vec{x})$ with $a(\vec{x}) \equiv \int d^3x' \nabla' \cdot \vec{A}(x')/|\vec{x} - \vec{x}'|$. If the scalar $a(\vec{x})$ is single-valued everywhere on $C$, then it is obvious that $\Phi(\vec{A}) = \Phi(\vec{A}_P)$! In other words, the physically measured geometric phase $\Phi(A_P)$ is dependent only on the physical projection $A_P$ of the gauge potential, and is quite independent of the pure gauge part dependent on $a(x)$.

### 5. Conclusions

We saw in the last section that physical effects stemming from nontrivial configuration spaces of test charges, like the Aharonov–Bohm effect, actually reinforce our contention that the gauge-invariant projection of the 4-vector potential plays the key role at the expense of the pure gauge piece. This approach completely demystifies the topic of gauge ambiguity, and champions special relativity through a totally Lorentz-covariant approach, free of gauge ambiguities.

Recently, it has been shown [8] that any *non-Abelian* gauge theory (with matter interactions) is classically equivalent to a set of Abelian gauge fields, whose self-interaction and interaction with matter are generated by a process of iterative Nöther coupling, without invoking the minimal coupling prescription. Since Abelian gauge fields are completely described by their physical projection, as elaborated in this paper, a mathematically simpler description of non-Abelian gauge fields, avoiding

any Faddeev–Popov gauge fixing, can be envisaged using our results. A preliminary attempt in this direction has been made in [6]. We hope to report more complete results elsewhere.

A related issue is that our approach can avoid the conundrum discussed many years ago by Gupta and Bleuler [10], associated with canonical quantization of the free Maxwell field, when the gauge potential is gauge fixed by means of a Lorentz-invariant gauge condition like the Lorenz–Landau gauge. Due to the indefinite spacetime metric, states in the Fock space of the theory are seen to possess negative norm. Gupta and Bleuler proposed that these unphysical Fock space states must be eliminated by subsidiary conditions imposed on the Fock space. In our approach, the projected 4-potential is actually a *spacelike* 4-vector with vanishing projection along the two linearly independent null directions of Minkowski 4-spacetime. The physical subspace of polarizations is $\mathcal{R}_2$, so problems associated with the indefinite metric of Minkowski spacetime ought not to be of consequence. We hope to report on this in detail elsewhere.

We have also been recently informed that similar projection operators have been considered in some contemporary works on quantum field theory, e.g., [11–14]. Even earlier, it was apparently J. L. Synge who first proposed projection operators to project out the physical degrees of freedom [15] of the electromagnetic field interacting with test charges. However, the formulation given here is that of the authors of this paper. Earlier, it has also been extensively discussed in class lectures on Maxwell electrodynamics given by one of us (PM) since 2005.

**Author Contributions:** The authors contribute equally to this paper.

**Funding:** This research received no external funding.

**Acknowledgments:** One of us (P.M.) would like to thank J. Navarro-Salas for his immense help in submitting this paper to Symmetry and his subsequent advise regarding publication.

**Conflicts of Interest:** The authors declare no conflict of interest.

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
