# Peer review of "Maxwell Electrodynamics in Terms of Physical Potentials"

_symmetry, doi:10.3390/sym11070915_

Reviewer 1 Report

This article discusses a gauge-invariant formulation of Maxwell electrodynamics by introducing a projection operator, which projects the vector potential to its physical (transverse) degrees of freedom, and then the whole theory can be built only on these physical degrees of freedom.

Although this article provides a clear exposition on historical developments and the logic of formulating the gauge-invariant electrodynamics, it does not provide something really new. This kind of projection operator and gauge-invariant formulation of Maxwell electrodynamics (later even generalized to non-Abelian gauge theory) is well-known in the literature, and has been discussed in many places. For instance, the idea of formulating electrodynamics in terms of physical (transverse) degrees of freedom was discussed around Eq.(120) in the book “Topology and Geometry in Physics” by E. Bick and F. D. Steffen (Eds.), and the same projection operator introduced in this article can be found around Eq.(6.31) in the online note on quantum field theory by David Tong.

Therefore, I cannot recommend the publication of this article in Symmetry.

Author Response

Thank you for your comments, the following is point-by-point response:

Critique : This article discusses a gauge-invariant formulation of Maxwell electrodynamics by introducing a projection operator, which projects the vector potential to its physical (transverse) degrees of freedom, and then the whole theory can be built only on these physical degrees of freedom.
Response : This is a fair representation of our work.

Critique : Although this article provides a clear exposition on historical developments and the logic of formulating the gauge-invariant electrodynamics, it does not provide something really new. This kind of projection operator and gauge-invariant formulation of Maxwell electrodynamics (later even generalized to non-Abelian gauge theory) is well-known in the literature, and has been discussed in many places.
Response : Admittedly, the projection operator has indeed appeared in the literature before, as we have also stated in the paper. However, that the Lorentz-covariant projection operator exists already in the relativistic, classical Maxwell equations, when written out in terms of the 4-vector potential, has not been shown explicitly, in earlier literature, to the best of our knowledge. The existence of a physical potential which is also spacetime divergence-free without any choice of gauge, within Maxwell electrodynamics, though perhaps implicit in earlier literature, has never been articulated as in our paper.

Critique : For instance, the idea of formulating electrodynamics in terms of physical (transverse) degrees of freedom was discussed around Eq.(120) in the book Topology and Geometry in Physics by E. Bick and F. D. Ste en (Eds.), and the same projection operator introduced in this article can be found around Eq.(6.31) in the online note on quantum eld theory by David Tong.
Response : We have perused the cited edited volume, especially the erudite article on BRST Quantization by JWVan Holten. However, the context of this article is not classical electrodynamics. Nor is the projection operator appearing in eqn. (120) of this article, the covariant projection operator in Fourier space that already appears in the Maxwell equations for the 4-vector potential, as shown in our paper. Spatial transversality within a chosen gauge is di erent from spacetime transversality appearing in a completely gauge invariant manner, which is the proven property of the physical potential in our paper. The same remarks apply to the projection operator appearing in the Coulomb gauge for electrodynamics in eqn (6.31) of the online Note on Quantum Field Theory by David Tong - it is expressed in terms of the spatial momentum components rather than the 4-momenta as in our paper. As emphasized above, the main point of our paper, is that the original Maxwell equations for the 4-potential give a unique gauge invariant and spacetime transverse projected vector potential which is as physical as the field strengths.

Reviewer 2 Report

This nicely written article is about the fundamental problem of gauge symmetry in electromagnetism.

The matter is widely discussed and dealt with in so many textbooks on quantum field theory and gauge symmetry where the first chapter is usually devoted to the Lorentz transformation and the covariant formulation of the equations of electromagnetism. As such, we do not believe this paper brings any new material. However, the discussion could be useful to new comers to the field.

Reviewer 3 Report

The paper is well written. The introduction is pedagogical and very compelling. The main goal is to point out that a gauge invariant, and also relativistic covariant, formulation of classical Maxwell electrodynamics can be achieved by means of a natural projection operator. This helps to clarify the traditional meaning of gauge invariance and gauge redundancy from a very sound and geometrical approach (based on the interesting observation by J. L. Synge in [6], as stressed in section IIIB2). The paper is well-accompanied with two illustrative examples where the role of the gauge invariant projection plays an essential role.

For the above reasons, I strongly recommend publication of this paper in the special issue of Symmetry (Symmetries of electromagnetism).

The paper could be improved if the authors explain with more detail the arguments for the identification of (30) with the projection operator. However, I leave this option to the own criterium of the authors.
